# Optimization of Phenolic Content Extraction and Effects of Drying Treatments on Physicochemical Characteristics and Antioxidant Properties of Edible Mushroom *Pleurotus ostreatus* (Jacq.) P. Kumm (Oyster Mushroom)

**DOI:** 10.3390/antiox13121581

**Published:** 2024-12-23

**Authors:** Marcelo Villalobos-Pezos, Ociel Muñoz-Fariña, Kong Shun Ah-Hen, María Fernanda Garrido-Figueroa, Olga García-Figueroa, Alexandra González-Esparza, Luisbel González-Pérez de Medina, José Miguel Bastías-Montes

**Affiliations:** 1Faculty of Agricultural and Food Sciences, Graduate School, Austral University of Chile (UACh), Valdivia 509000, Chile; mvillalobospezos@gmail.com; 2Institute of Food Sciences and Technology, Austral University of Chile (UACh), Valdivia 509000, Chile; kshun@uach.cl (K.S.A.-H.); maria.garrido@alumnos.uach.cl (M.F.G.-F.); olga.garcia@uach.cl (O.G.-F.); alexandra.gonzalez@uach.cl (A.G.-E.); 3Laboratory of Biomaterials, Department of Chemical Engineering, Faculty of Engineering, University of Concepción (UdeC), Concepción 4030000, Chile; luisbgonzalez@udec.cl; 4Department of Food Engineering, University of Bío-Bío, Chillán 3780000, Chile; jobastias@ubiobio.cl

**Keywords:** edible mushroom, drying treatments, ultrasonic-assisted extraction, phenolic compounds, antioxidant activity, ergothioneine, *P. ostreatus*

## Abstract

Edible mushrooms have been part of the human diet for centuries. Traditionally, they have been used for culinary and medicinal purposes due to their chemical composition and nutritional value, including their high antioxidant activity attributed to key metabolites such as phenolic compounds and ergothioneine. *P. ostreatus* mushrooms, known for their potent antioxidant activity, are prone to spoilage shortly after harvest, making preservation methods necessary. Various drying methods were evaluated to determine their effects on physicochemical characteristics, antioxidant properties, and ergothioneine content. Mushrooms were subjected to freeze-drying (FD), hot-air-drying (HAD), and microwave-vacuum-drying (MVD). The rehydration rate, color, and microstructural characteristics of extracts from fresh and dried *P. ostreatus* mushrooms were evaluated. Additionally, the total soluble phenolic content and antioxidant activity were assessed using DPPH and ORAC assays, along with the determination of ergothioneine content. FD and HAD at 40 °C showed the best results regarding the physicochemical characteristics. In terms of total soluble phenolic content, antioxidant activity, and ergothioneine content, the mushrooms dried using HAD at 40 °C showed the best retention of bioactive compounds. Among the evaluated methods, HAD at 40 °C proved to be a suitable alternative for processing *P. ostreatus*.

## 1. Introduction

Edible mushrooms have been part of the human diet for centuries, primarily due to the variety of flavors they can provide. In many Asian countries, mushrooms are traditionally used in both food and medicine, and they are useful for preventing chronic diseases due to their chemical composition [1]. Edible mushrooms are also nutritionally desirable because of their low energy value, high fiber content, and high antioxidant capacity [2]. The antioxidant capacity is exerted through potent radical scavengers and reducing agents, and it is primarily attributed to the phenolic content [3] and to the combination of phenolic content with ergothioneine [4]. Both wild and cultivated mushrooms serve as the primary source of this molecule [5], which is documented as an excellent antioxidant [6].

Paul Kummer officially identified the genus *Pleurotus* in 1871. These edible mushrooms, known as oyster mushrooms, are the third most extensively cultivated edible mushroom species worldwide, followed by the genera *Agaricus* and *Lentinula*. They exhibit a short life cycle, high adaptability, and reported health benefits [7]. Oyster mushrooms are classified as basidiomycetes. Their natural habitat comprises tropical and subtropical forests, where they thrive on dead trunks, logs, and stumps of deciduous trees [8]. As a saprophytic mushroom, *P. ostreatus* obtains its nutrients from dead organic matter [9] and can be cultivated on various substrates prepared from lignocellulosic waste [8], which are largely produced as byproducts in the agricultural, forestry, and food processing industries [10]. The reuse of waste adds value to the production of edible mushrooms [11].

Oyster mushrooms are highly perishable and begin to deteriorate immediately after harvest [12]. They are more delicate and sensitive than mushrooms from the genus *Agaricus* [11], which are also highly perishable, with a shelf life of only about 24 h under environmental conditions [13]. Once deteriorated, *P. ostreatus* mushrooms develop an unfavorable taste for direct consumption [14]. Fresh mushrooms must be processed to extend their shelf life for out-of-season consumption, with drying being one of the most common methods used to reduce the moisture content of oyster mushrooms [12], maintaining a taste like that of fresh mushrooms when rehydrated [14], although a series of changes may occur that would affect their nutritional value [15]. Therefore, rapid and efficient drying processes that can control nutritional loss in edible mushrooms have gained interest. Freeze-drying is an expensive method, yet it yields dried foods with low nutritional degradation, in contrast to the results of thermal treatment methods [13]. Hot-air-drying requires a prolonged drying time due to low thermal conductivity [16], while microwave-vacuum-drying combines the rapid energy transfer of microwave drying with the rapid mass transfer at low temperatures under vacuum conditions [17], reducing drying time by 70% and 90% compared to that of hot-air-drying and freeze-drying systems respectively [18]. Owing to the combined advantages of microwave-vacuum-drying, less energy is consumed in the process [19]. It has been used on edible mushrooms, resulting in high-quality dried products compared to those from hot-air-drying [17,20]. According to available evidence, to date, microwave-vacuum-drying has not been tested on *P. ostreatus* mushrooms.

The demand for new functional ingredients or bioactive compounds of natural origin is increasing and has, in recent years, promoted a growing interest in the extraction of bioactive compounds from foods such as mushrooms [21]. Bioactive compounds from mushrooms are valuable resources, and extraction methods are vital. The most conventional methods are time-consuming and require considerable volumes of organic solvents, which can generate higher disposal costs and environmental issues. There is a growing demand for more environmentally friendly alternatives that require shorter extraction times and lower amounts of organic solvents [7] for determining total soluble phenolic, as well as ergothioneine levels [22]. Ultrasonic-assisted extraction is much more economical in terms of time, solvent usage, and energy [23] and is considered one of the most promising extraction technologies for biologically active substances [24].

The aim of this research was to determine the optimal conditions for the variables of temperature, time, and solvent for the maximization of the total soluble phenolic extraction of *P. ostreatus* through a response surface model (RSM). Additionally, a comparison of three drying processes, namely freeze-drying, hot-air-drying, and microwave-vacuum-drying, was performed in regards to the rehydration rate, color properties, and microstructural properties of *P. ostreatus* mushrooms, as well as the total soluble phenolic content, antioxidant activity, and ergothioneine content of the extracts.

## 2. Materials and Methods

### 2.1. P. ostreatus Mushroom Samples

The fruit bodies of the fungus *P. ostreatus* were obtained from local supermarkets in the city of Valdivia, Chile. A portion of the purchased mushrooms were immediately subjected to the extraction procedure in their fresh state, while those that underwent drying treatments were stored in a desiccator until analysis.

### 2.2. Drying Processes

Before drying, the *P. ostreatus* mushrooms were selected and homogenized by size. The mushrooms, cut in half, were dried in three different ways, i.e., through freeze-drying (FD), hot-air-drying (HAD), and microwave-vacuum-drying (MVD). In each drying experiment, the mushrooms were dried on Petri dishes, starting with a sample of 20 ± 1 g of mushrooms. The drying process was carried out until a constant weight was achieved. The moisture content was determined to establish the percentage of moisture in the dried mushroom for each treatment. The final moisture content of the mushrooms dried by freeze-drying, hot-air-drying, and microwave-vacuum-drying was 6.24 ± 0.05%, 7.27 ± 0.52%, and 6.64 ± 0.32%, respectively. The assays were conducted in triplicate.

#### 2.2.1. Freeze-Drying

Freeze-drying was carried out as described by Pino et al. [25]. The samples were dried in a freeze dryer (Beta 1–8 LDplus, Christ, Osterode am Harz, Germany) at a pressure of 0.021 mbar and a condenser temperature of −55 °C until constant weight was achieved. The weight loss in the freeze-drying process was determined using a weighing system installed in the vacuum chamber, monitored with an Arduino Uno microcontroller and coupled between the freeze dryer and a computer.

#### 2.2.2. Hot-Air-Drying

The hot-air-drying process was carried out as described by Apati et al. [11], with some modifications. The samples were dried in an air flow oven (ZRD-A5055, Zhicheng, China) at temperatures of 40, 60, and 80 °C. The samples were weighed every 10 min using a digital electronic scale to determine the drying kinetics until constant weight was achieved.

#### 2.2.3. Microwave-Vacuum-Drying

The microwave-vacuum-drying process was carried out according to the methods of Kantrong, et al. [26], with some modifications. The samples were dried in a microwave-vacuum oven (ZYWX—1KW, Zhengshou Keda Machinery and Instrument Equipment Co., Zhengzhou, Henan, China) at microwave powers of 160, 210, and 260 W at an absolute pressure of 20 kPa. The samples were weighed every 2 min using a digital electronic scale to determine the drying kinetics until constant weight was achieved.

### 2.3. Color Measurement 

The color measurement of the fresh and dried *P. ostreatus* samples was determined as described by Hou et al. [27]. The color parameters of the samples were evaluated by reflectance with a colorimeter (CR-400, Minolta Chroma Meter, Tokyo, Japan) that was previously calibrated with a blank. The results were expressed as CIELAB values. The assay was performed in triplicate.

### 2.4. Rehydration Rate of Dried P. ostreatus Mushrooms

The rehydration rate (Rf) was determined following the procedures described by Qiu et al. [28], with some modifications. During the rehydration process, the samples were removed from the water bath at intervals of 30 s up to 2 min, then every 1 min until 6 min, and then every 2 min until constant weight was reached. The samples were weighed after draining on filter paper using a digital electronic scale. Rf was calculated using the following formula: Rf = Gf/Gg, where Gf is the weight of the rehydrated and drained sample, and Gg is the initial dry sample weight. To determine the nature of the diffusion inside the mushrooms, the semi-empirical equation developed by Peppas [29] was used, as follows: α = (m_t_/m_∞_) = kt^n^. Where m_t_ and m_∞_ are the absorbed mass at time t and at equilibrium, respectively, k is the characteristic constant related to the lattice structure of the mushrooms, and n is the diffusion exponent. The description of the penetration mechanism was provided for m_t_/m_∞_ ≤ 0.6 [30]. The procedures were conducted in triplicate.

### 2.5. Microstructure of Dried P. ostreatus Mushrooms

A scanning electron microscope (SEM) (Carl Zeiss, EVO, Cambridge, UK) was used to visualize the microstructure of the stipe (stem, peduncle) of the dried mushrooms. The selected and cut samples were mounted on carbon adhesive tape on slides, adding bridges of the same material to improve electrical conductivity, and were then coated with gold/palladium using a sputter coater (Leica Microsystems GmbH, EM ACE200, Wetzlar, Germany).

### 2.6. Ultrasound-Assisted Water Bath Extraction of P. ostreatus

The extraction method was performed according to the methods of Wang, et al. [23], with some modifications. A total of 2 g of raw mushrooms was weighed for optimizing the extraction of total soluble phenolic content, and 4 g of raw mushrooms and 0.4 g of dried mushrooms were used for comparing the effects of the drying treatments. The raw mushrooms were ground and homogenized in an immersion blender (2616, Oster, China), and the dried mushrooms were powdered in a coffee grinder (Sindelen, MOL—165, La Florida, Santiago, Chile). They were weighed in a 50 mL centrifuge tube. Subsequently, 15 mL of an aqueous ethanolic solvent (in different ratios of *v*/*v* ethanol/distilled water of 100/0, 50/50, and 0/100 in each assay) was added, manually homogenized, and extracted in an ultrasonic water bath at 67 ± 2 °C (Neytech, Ultrasonik, 57H, Yucaipa, CA, USA); at the end of each extraction time, the 50 mL tubes were centrifuged at 3000 g for 10 min (Rotofix 32A, Hettich, Tuttlingen, Germany), transferring the supernatant to another 50 mL centrifuge tube and resuspending the precipitate with the solvent again, completing three extraction cycles.

### 2.7. Optimization of the Extraction Procedure Using Response Surface Methodology for Fresh P. ostreatus Extracts

Response surface methodology was applied to investigate the effect of three independent variables, i.e., the solvent at three different ratios of ethanol/distilled water (100/0, 50/50, and 0/100 *v*/*v*), three different extraction times (30, 60, and 90 min), and three different extraction temperatures (50, 65, and 80 °C), on the dependent variable, total soluble phenolic content, in *P. ostreatus* extracts. The experiments were conducted using a three-level factorial design in three blocks, utilizing Statgraphics Plus v.5.1 software. The complete design consisted of 30 experimental runs, including one central point per block, and these were performed in triplicate.

### 2.8. Determination of Total Soluble Phenolic Content in Fresh and Dried Mushroom Extracts

The total soluble phenolic content of the extracts was measured using the Folin–Ciocalteu method [31]. A volume of 40 µL of extract sample was diluted in 3160 µL of distilled water and mixed with 200 µL of Folin–Ciocalteu reagent. The mixture was vortexed for 7–10 s, then 600 µL of 20% sodium carbonate (Na_2_CO_3_·5H_2_O) was added, and it was vortexed again for 7–10 s. The reaction developed in the dark, protected from sunlight, for 120 min at room temperature. Absorbance was measured at 765 nm in a spectrophotometer (Spectronic, Genesys 5, Rochester, NY, USA). Quantification was performed against a gallic acid calibration curve. The results of the total soluble phenolic content were expressed as mg of gallic acid equivalent per gram of dry mass (mg GAE/g of d.m.). All measurements were performed in triplicate.

### 2.9. Determination of Antioxidant Activity of Fresh and Dried Mushroom Extracts

#### 2.9.1. DPPH Radical Scavenging Activity

The radical scavenging activity of the extracts from the samples was determined through a 2,2-diphenyl-1-picrylhydrazyl (DPPH) assay, according to the method of Brand et al. [32]. From an aliquot of the sample extract previously centrifuged at 12,000 rpm for 10 min, 100 µL was added to 2.9 mL of 1 mM DPPH solution in methanol in a glass cuvette and mixed completely for 10 s. The absorbance of the samples was measured at 515 nm using a spectrophotometer (Spectronic, Genesys 5, Rochester, NY, USA). The results were expressed as mmol Trolox Equivalent per 100 g of dry mass (mmol TE/100 g d.m.). The assay was conducted in triplicate.

#### 2.9.2. Oxygen Radical Absorption Capacity (ORAC) Assay

The capacity of the extracts to delay the oxidative decomposition of fluorescein induced by 2,2′-azobis(2-methylpropionamidine) dihydrochloride (AAPH) was measured using the method of Ou et al. [33]. A previously prepared aliquot of the sample extract was centrifuged at 12,000 rpm for 10 min. Then, these samples of the extract and reagents were prepared and diluted in 75 mM phosphate buffer (pH 7.4) in a black microplate with a flat and transparent bottom. Volumes of 45 μL of the sample and 175 μL of fluorescein at 108 mM were deposited. This mixture was preincubated for 30 min at 37 °C; next, 50 μL of AAPH solution at 108 nM was added. The microplate was immediately placed in the dual-scanning microplate spectrofluorometer (Molecular Devices, Gemini XPS, Sunnyvale, CA, USA) for 60 min using a 538 nm emission and a 485 nm excitation wavelength. The fluorescence readings were recorded every 3 min, and the microplate was automatically shaken before and after each reading.

The microplate was immediately placed in the double-scanning microplate spectrofluorometer for 60 min, with excitation and emission wavelengths set at 485 nm and 538 nm, respectively. The fluorescence readings were recorded every 3 min. The microplate was automatically shaken before and after each reading. Trolox at 6, 12, 18, and 24 μM was employed as a standard to obtain the calibration curve, and the buffer was used as a blank. ORAC values were calculated as the area under the curve (AUC) values, and these were compared with the Trolox calibration curve. The results were expressed as mmol Trolox Equivalent per 100 g of dry mass (mmol TE/100 d.m.). The assay was performed in triplicate.

### 2.10. Ergothioneine Content by HPLC of Fresh and Dried Mushroom Extracts

The determination of ergothioneine content in the extracts was performed according to the method described by Dubost et al. [6]. High-performance liquid chromatography (HPLC) was used, employing a ternary pump (L-6200, Merck-Hitachi, Darmstadt, Germany), a UV-visible detector (L-4250, Merck-Hitachi, Darmstadt, Germany), and an autosampler (717 Plus, Waters, Milford, MA, USA). The chromatograms were analyzed using Clarity software (DataApex, Prague, Czech Republic). In summary, the chromatographic separation of ergothioneine was achieved using two reversed-phase columns (C18, 5 μm, 250 mm × 4.6 mm, GL Sciences, InertSustain, Tokyo, Japan) connected in tandem. The mobile phase was prepared as an isocratic solution consisting of 50 mM sodium phosphate buffer, 3% acetonitrile, and 0.1% trimethylamine, adjusted to pH 7.3, with a flow rate of 1 mL/min. A wavelength of 254 nm was used for monitoring. The injection volume was 20 μL, with a column temperature of 30 °C. The identification and quantification of ergothioneine were confirmed by comparing the retention times and peak areas of the sample extracts with those obtained from different concentrations of the ergothioneine standard. The results were expressed as mg of ergothioneine per gram of dry mass (mg ergothioneine/g d.m.). The assay was performed in triplicate.

### 2.11. Statistical Analysis

All assays were performed in triplicate. The results were reported as mean ± standard deviation. The analysis of variance and significant differences were evaluated using Simple ANOVA with Statgraphics Plus v. 5.1 software (Statistical Graphics Corp., Herndon, VA, USA). The Tukey multiple range test included in the statistical program was also used to demonstrate the existence of homogeneous groups within each of the parameters at a 95% confidence interval. Data were plotted using OriginPro 10.1.5.132^®^ software (OriginLab Corporation, Northampton, MA, USA).

## 3. Results and Discussion

### 3.1. Drying Processes Applied to the Edible Mushroom P. ostreatus

The variations in the moisture ratios of the mushrooms *P. ostreatus* as a function of time during freeze-drying, hot-air-drying, and microwave-vacuum-drying are presented in Figure 1a,b. The longest drying processes corresponded to freeze-drying (>700 min) and hot-air-drying at 40 °C (500 min). For hot-air-drying, as expected, a decrease in drying time of 48% was observed with an increase in temperature from 40 to 60 °C and 15.4% when the temperature increased from 60 to 80 °C. These results are consistent with those from previous research on the drying of *P. ostreatus*, showing a reduction in processing time close to 40% when the drying temperature increased from 40 to 60 °C [11]. The shortest drying time was achieved at 80 °C (220 min) (Figure 1a). For microwave-vacuum-drying (Figure 1b), the time required to reduce the moisture content to a constant weight depends on the microwave power level [17]. Due to the preferential absorption of microwave energy by water molecules [20] and higher power, more heat is generated inside the sample [26]. Vacuum pressure does not significantly influence drying time as much as does microwave power [17]. For this study, a vacuum pressure of 20 kPa was selected. Similarly, consistent with previous studies on edible mushrooms, a shorter drying time (30 min) was observed under microwave power conditions of 260 W at 20 kPa (Figure 1b), as higher microwave power and lower vacuum pressure promote greater moisture removal and effective dehydration, reducing drying time [26]. Comparing the shortest drying times between hot-air and microwave-vacuum-drying, there was an 86% reduction in drying time, aligning with the results of previous studies regarding hot-air and microwave-vacuum-drying processes [18]. In Figure 2, macroscopic images of the mushrooms after the different drying methods can be observed.

### 3.2. Color Measurement of Fresh and Dried Mushrooms

The most popular methods of measuring color involve instruments that measure surface reflectance. In the CIELAB coordinates, the values L*, a*, and b* describe a three-dimensional color space. The value L* is the vertical axis and defines brightness, while the values a* and b* are perpendicular horizontal axes that define the spectrum from red to green and blue to yellow, respectively [34]. Color is an essential attribute that influences the acceptability of a product by consumers [13] and is one of the quality criteria of a dried product, whose modification is due to the degradation of pigments and enzymatic darkening during drying treatments [35]. The drying processes had a significant impact on the color parameters of *P. ostreatus*. Table 1 lists the L*, a*, b*, and ΔE values (=ΔL^2^ + Δa^2^ + Δb^2^)^½^, which represent the colorimetric difference between the sample and the standard white reflecting plate [36]. The L*, a*, b*, and ΔE values for fresh *P. ostreatus* mushrooms were 51.23, −0.86, 11.32, and 52.48, respectively. Visually, freeze-dried oyster mushrooms have the same color as fresh mushrooms, which can be explained by the low concentration of oxygen present in the vacuum conditions of freeze-drying and consequently, the lower activity of enzymatic reactions, which are the main cause of discoloration in dried mushrooms [35].

The L*, a*, b*, and ΔE values for freeze-dried *P. ostreatus* mushrooms are slightly different compared to those of other drying conditions. In general, the L* value decreases after drying treatment compared to that of fresh mushrooms, and products with a higher L* value are preferred by consumers [37]. The highest L* value was observed for the freeze-dried samples, followed by the hot-air-dried samples at 40 °C and the microwave-vacuum-dried samples at 260 W/20 kPa. The lowest L* value was observed for the hot-air-dried oyster mushrooms at 80 °C, where darkening occurred due to the Maillard reaction, favored by the high drying temperature (Figure 3). In general, for mushrooms dried with hot air, the L* value decreases with increasing temperature because heat promotes the Maillard reaction [38] and causes more pigmentation in the mushrooms [12], while for microwave-vacuum-dried mushrooms, the shorter drying time, due to the increased microwave power and reduced vacuum pressure, combined with a lower drying temperature, results in a higher L* value [26]. No significant difference was observed between the L* values of mushrooms dried by hot air at 40 °C and mushrooms dried by microwave-vacuum at 260 W/20 kPa.

The a* values of the fresh and freeze-dried mushrooms showed only slight but not significant differences; the highest a* value was observed for microwave-vacuum-dried mushrooms at 160 W/20 kPa, and the lowest was observed for samples dried at 260 W/20 kPa, followed by those dried at 210 W/20 kPa. The hot-air-dried mushrooms dried at 80 °C showed the highest a* value, which could be related to greater pigmentation compared to that for the other drying treatments [36]. For the b* value, the lowest value was observed in mushrooms dried under microwave-vacuum conditions at 260 W/20 kPa, while the highest b* values were determined in the freeze-dried mushrooms, the hot-air-dried samples at 40 °C, and in the microwave-vacuum-dried samples at 160 W/20 kPa (Table 1). The L*, a*, and b* values of mushrooms dried by microwave-vacuum at 260 W/20 kPa indicate a rapid reduction in moisture content, making them less susceptible to pigmentation changes due to enzymatic or Maillard reactions. The ΔE values are listed as follows: fresh > FD > HAD 40 °C > MVD 160 W > MVD 260 W > MVD 210 W > HAD 60 °C > HAD 80 °C. The significant differences between the microwave-vacuum-dried mushrooms are minimal; for the hot-air-dried mushrooms, the ΔE value decreases as the drying temperature increases, while only a slight significant difference was observed between the freeze-dried and fresh mushrooms. For the edible mushroom *Lentinus edodes*, the highest L* value and the lowest color difference (ΔE) are used as a reference for color quality [20]. According to this criterion, the mushrooms *P. ostreatus* dried by freeze-drying, hot-air drying at 40 °C, and microwave-vacuum-drying at 260 W/20 kPa are those that have the highest L* values and the lowest color difference (ΔE).

### 3.3. Microstructure of Fresh and Dried Mushrooms

The influence of drying on the quality of dried mushrooms was studied by scanning electron microscopy (SEM). SEM images of the internal and surface structures of freeze-dried *P. ostreatus* mushrooms (Figure 3), hot-air-dried mushrooms (Figure 4 and Figure 5), and microwave-vacuum-dried mushrooms are shown in Figure 6 and Figure 7. The drying processes caused very different changes in the microstructural properties of the dried *P. ostreatus* mushrooms. Freeze-drying produced a high-quality product [39], with a characteristic structure that is more porous both internally (Figure 3a,b) and superficially (Figure 3c,d), attributes resulting from the low drying temperature. For hot-air-dried mushrooms, it has been shown that this drying process, in contrast to the results for microwave-vacuum-drying, results in a much less-open structure and pores [17]. On the other hand, for *P. ostreatus*, it was observed that the effect of drying processes on the structure and pores depends on the conditions of the drying process. For mushrooms dried at 40 °C (Figure 4a,b), a more preserved structure and pores were observed compared to mushrooms dried at 60 °C (Figure 4c,d) or 80 °C (Figure 4e,f), where a collapse of the structure occurred. Regarding the surfaces of hot-air-dried mushrooms (Figure 5a,f), greater porosity and less shrinkage were noticeable when they were dried at 40 °C (Figure 5a,b) compared to 60 °C (Figure 5c,d) or 80 °C (Figure 5e,f), for which greater contraction and roughness were evident. In other studies, after applying drying treatments to *P. ostreatus*, the microstructures of the mushrooms were examined by X-ray microtomography analysis, revealing that the freeze-drying, hot-air-drying, and microwave-drying processes produce heterogeneous structures and pore numbers in the mushrooms. There were also significantly high values for the degree of anisotropy, which is the absence of a preferred orientation of pores in the structure for *P. ostreatus* mushrooms dried mainly by microwave, as high temperatures during microwave drying lead to an increase in water release, resulting in significant deformation in the structure of the dried mushrooms [35]. By including vacuum pressure at 20 kPa for the three microwave powers, the combined effect of microwave power and vacuum pressure on the structure and pores could be observed. Microwave-vacuum-drying, compared to freeze-drying and hot-air-drying, resulted in a faster and greater release of water from the matrix, as a consequence of vacuum drying [40] and a decreased in the boiling temperature of water, which correlated well with the moisture contents achieved after the drying processes. In microwave-vacuum-drying at a power of 210 W/20 kPa (Figure 6c,d), a larger and better-preserved structure and pores were observed compared to the results for the treatment at 160 W/20 kPa (Figure 6a,b). A collapse of the structure was observed at 260 W/20 kPa (Figure 6e,f), possibly due to the speed of moisture transfer from the mushroom to the vacuum chamber [26]. At all three microwave powers, contraction, misalignment, and shrinkage of the surface can be observed (Figure 7a,f), particularly at 260 W/20 kPa (Figure 7e,f). The microstructural differences observed in *P. ostreatus* after the application of hot-air-drying and microwave-vacuum-drying treatments differ from those in the previous research on edible mushrooms, as microwave-vacuum-drying shows better preservation of the microstructural characteristics [17,20]. However, shrinkage and collapse of the structure can also be observed in microwave-vacuum-dried edible mushrooms [40,41] as observed in the mushroom *P. ostreatus* (Figure 6 and Figure 7).

### 3.4. Rehydration Rate

Dried mushrooms can be rehydrated by immersion in water before consumption. Rehydration characteristics are used as a quality parameter [40] and indicate whether physical and chemical changes occurred during the drying process [11]. The rehydration rate represents a way to measure the structural damage resulting from the drying process. Figure 8 shows the rehydration capacity of dried mushrooms. Freeze-dried mushrooms have a greater rehydration capacity compared to those of other drying processes, which agrees with the results of previous studies on *P. ostreatus* [35], explained by the previously mentioned structural differences. Generally, hot-air-drying processes have significant disadvantages concerning rehydration properties [26]. However, for *P. ostreatus*, the rehydration rate is listed as follows: FD > HAD 40 °C > HAD 60 °C > MVD 210 W > MVD 160 W > MVD 260 W > HAD 80 °C. For hot-air-drying, the increase in temperature had a negative influence on the rehydration capacity of the mushrooms [11]. For microwave-vacuum-dried edible mushrooms, previous studies have shown that rehydration properties improved under low vacuum pressure and high microwave power, while the rehydration rate mainly depends on vacuum pressure [17]. This would lead to a more porous structure, as well as a greater pressure difference between the vacuum chamber and the internal pressure in the food, resulting in a less dense but more expanded and fluffy structure, subsequently achieving a greater water absorption capacity [26]. It can be seen in Figure 8 that at different microwave powers (160–260 W) under a vacuum pressure of 20 kPa, the best rehydration capacity was achieved at 210 W/20 kPa, showing an appropriate relationship between microwave power and vacuum pressure. As observed in the SEM images, drying at 260 W/20 kPa caused a structural collapse that affected the rehydration capacity, similar to that for hot-air-drying at 80 °C, where a structural collapse was also observed, presumably due to the temperature increase during microwave-vacuum-drying, especially towards the end of the drying treatment. This affects the physicochemical properties as observed in the rehydration rate [42]. Table 2 presents the fitting of the rehydration data to the mathematical model of Peppas [29]. It can be observed that the predominant rehydration mechanism is the fiber relaxation over pore diffusion, except at 210 W/20 kPa (*n* values greater than 0.5), indicating the coexistence of both mechanisms. However, at 260 W/20 kPa, which exhibits a value very close to 0.5, pore diffusion predominates. These results establish a direct relationship between the increase in microwave power and vacuum pressure and the predominance of pore diffusion, which is a variable that can be manipulated for specific commercial purposes. The k values are evidence of structural changes in the mushroom matrix, correlating with the results of the SEM images and color parameters.

### 3.5. Effect of Different Extraction Factors in Ultrasonic-Assisted Water Bath on the Total Soluble Phenolic Content of P. ostreatus Extracts

Temperature and time were controlled during the extraction procedures. The water bath temperature was maintained as constant as possible throughout the extraction process. The process variables were set at ratios of 100/0, 50/50, and 0/100 (*v*/*v*) for the solvent mixture of ethanol/distilled water; the extraction time was set at 30, 60, and 90 min; and the ultrasonic bath temperature was set at 50, 65, and 80 ± 2 °C. As shown in Table 3 and Table 4, the maximum extraction of the total soluble phenolic content of fresh *P. ostreatus* was significantly dependent on the experimental extraction temperature (X_1_) and solvent (X_3_) factors (*p* < 0.05). The quadratic parameter X_3_^2^ was also significant (*p* < 0.05). There is limited evidence for variable optimization for maximizing the extraction of phenolic content in *P. ostreatus* [22]. The ultrasonic-assisted extraction method could ensure the maximization of extraction due to mechanical effects that facilitate greater penetration, improving solvent transfer into the matrix cells. The induced cavitation effect caused cell wall rupture and the release of their contents into the solvent, leading to higher yields in shorter amounts of time at lower processing temperatures [43]. Therefore, these results could contribute to knowledge regarding methods for extracting the phenolic content of oyster mushrooms.

### 3.6. Response Surface Methodology

Figure 9a–c show the interactions between temperature (X_1_), time (X_2_), and solvent (X_3_), as well as their combined effect on the extraction of total soluble phenolic content (Y) from fresh *P. ostreatus* mushrooms. Figure 9a shows the interaction between temperature (X_1_) and time (X_2_). Within the range of 65–70 °C, and independent on time (X_2_), higher values of phenolic content were obtained. Figure 9b illustrates the interaction between time (X_2_) and solvent (X_3_), and similar to the results in Figure 9a, higher values of phenolic content were observed when distilled water constituted 100% of the solvent, independent on time. Figure 9c demonstrates the interaction between temperature (X_1_) and solvent (X_3_). It can be seen that within the temperature range of 65–70 °C and with a proportion of 100% distilled water in the solvent, the phenolic content achieved higher extraction values. The optimal conditions for temperature, time, and solvent in order to maximize the extraction of phenolic compounds from *P. ostreatus* were determined to be 67 °C for temperature, 90 min for time, and 100% distilled water as the solvent. These extraction conditions for the three variables yielded the results discussed below regarding total soluble phenolic content, antioxidant activity, and ergothioneine. These results are consistent with recent research findings that the aqueous extracts of *P. ostreatus* achieve the highest phenolic content [44].

### 3.7. Determination of Total Soluble Phenolic Content in Aqueous Extracts of Fresh and Dried P. ostreatus Mushrooms

Phenolic compounds are among the most important metabolites of fungi with antioxidant properties [35], and those found in *Pleurotus* mushrooms have been shown to be powerful [22], similar to those of other edible mushrooms, in terms of phenolic content and antioxidant activity, with recognized therapeutic potential [45]. Drying processes led to significant changes in the extraction of phenolic compounds from the mushroom *P. ostreatus*, depending on the drying process applied, which can be ranked as follows: fresh > HAD 80 °C > HAD 40 °C > HAD 60 °C > FD > MVD 260 W > MVD 160 W > MVD 210 W (Figure 10). The highest total phenolic content was observed in the aqueous extracts from fresh mushrooms (25.38 mg GAE/g d.w.), followed by the aqueous extracts from mushrooms dried by hot air at 80 °C (20.07 mg GAE/g d.w.) and those from mushrooms dried by a microwave-vacuum at 160 W/20 kPa and 210 W/20 kPa (14.58 mg GAE/g d.w. and 14.49 mg GAE/g d.w., respectively). The phenolic content observed in the aqueous extracts of freeze-dried *P. ostreatus* mushrooms may have been influenced by the process’s inability to inactivate oxidative enzymes due to low temperature conditions [35], as indicated by the better conservation of structure seen in SEM images, maintaining their cellular components. In hot-air-drying, phenolic compounds are more susceptible to drying time than to temperature [15], with a significantly greater retention of phenolic compounds in aqueous extracts of mushrooms dried at 80 °C compared to those dried at 40 and 60 °C, possibly influenced by the inactivation of polyphenol oxidase that occurs above 60 °C [46] and by the structural collapse observed in the SEM images, allowing phenolic compounds to be released from the matrix [35,38]. The lower content observed in extracts from hot-air-dried mushrooms compared to fresh mushroom extracts is consistent with previous research on *P. ostreatus* [47]. In the aqueous extracts from mushrooms dried at 260 W/20 kPa, the structural collapse seen in the SEM images facilitated the degradation of cellular components and the release of phenolic content from the matrices, making them more accessible to extraction [35], compared to the results for extracts from mushrooms dried at 160 W/20 kPa and 210 W/20 kPa, which show better-preserved structures in the SEM images. The extraction of phenolic compounds during the extraction process may have been influenced by the quality of the cellular structures after the drying process, as the extraction depends on the solvent’s penetration into the matrix and the diffusion of phenolic compounds out of the matrix, as well as the dissolution of solutes from the matrix into the solvent phase [7]. The phenolic content could determine the antioxidant activity discussed subsequently, since a good correlation between phenolic content and antioxidant activity has been observed in edible mushrooms [48].

### 3.8. Antioxidant Activity of Aqueous Extracts from Fresh and Dried P. ostreatus Mushrooms

The drying processes resulted in significant changes in the antioxidant activity, with changes depending on the drying method applied to the *P. ostreatus* mushroom. For the antioxidant activity in the DPPH assay (Figure 11a), significant differences were observed between the aqueous extract of fresh mushrooms (2.98 mmol TE/100 g d.m.) and the aqueous extracts from mushrooms dried by hot-air at 60 °C or by freeze-drying (1.30 mmol TE/100 g d.m. and 1.21 mmol TE/100 g d.m.). For the antioxidant activity in the ORAC assay (Figure 11b), significant differences were also observed between the fresh mushroom extract (32.54 mmoles TE/100 g d.m.), and the extracts from mushrooms dried with hot-air at 40 °C and by microwave-vacuum at 210 W/20 kPa (22.12 mmoles TE/100 g d.m. and 13.87 mmoles TE/100 g d.m. respectively). The phenolic content of edible mushrooms serves as one of the most effective indicators of antioxidants [48] and is the main carrier of these substances; they can donate electrons and hydrogen atoms, and it was observed that aqueous extracts of *P. ostreatus* could scavenge DPPH radicals and absorb oxygen radicals (ORAC) [24]. The trend observed between total soluble phenolic content and antioxidant activity through the DPPH and ORAC assays of aqueous extracts of *P. ostreatus* is consistent with the results of previous research on extracts of edible mushrooms, in which a higher amount of phenolic content has been associated with increased antioxidant activity [6,49,50]. In particular, the results obtained with the aqueous extracts of *P. ostreatus* dried by hot-air at 80 °C in the DPPH assay (Figure 11a) can be attributed to the action of other compounds in addition to the phenolic content [50]. In the case of the ORAC assay in general, the trend observed between the total soluble phenolic content (Figure 10) and the assay results (Figure 11b) is in line with the results of research on edible mushrooms, as a good correlation has generally been found [48]. It is important to mention that the antioxidant activity of *P. ostreatus* extracts can be attributed to the presence of many bioactive compounds, including flavonoids, phenols, and peptides [45], given the complexity of the chemical composition of *P. ostreatus* mushrooms [24]. It has even been observed that each compound within the phenolic content may exhibit differing antioxidant activity [51]. Nevertheless, a significant contribution of the phenolic content to the antioxidant properties has been observed in edible mushrooms, including *P. ostreatus*, which could also explain the relationship between phenolic content and the antioxidant activity [49].

### 3.9. Content of Ergothioneine by HPLC in Aqueous Extracts of Fresh and Dried P. ostreatus Mushrooms

The ergothioneine content in *P. ostreatus* mushrooms, compared to other cultivated mushroom species, is high [6,52], and it is considered an excellent source of ergothioneine, as is the case for other edible mushroom species [4]. The ergothioneine content in dried mushrooms varies according to the applied drying processes due to their influence on the structure of the tissues and cells. The drying processes break down proteins into amino acids that are precursors to the synthesis of ergothioneine. In extracts of *Pleurotus* mushrooms subjected to drying processes, only natural-air-drying promotes an increase in ergothioneine content, while hot-air-drying at 60 °C resulted in a significant decrease [53]. In this study, the ergothioneine content (Figure 12) in aqueous extracts shows significant differences between the aqueous extract of fresh mushrooms (6.52 ± 0.16 mg/g d.m.) and the aqueous extract of mushrooms dried by microwave-vacuum at 260 W and 20 kPa (4.99 ± 0.31 mg/g d.m.), by hot-air at 60 °C (3.95 ± 0.10 mg/g d.m.), or by freeze-drying (2.63 ± 0.02 mg/g d.m.). This can be ranked as follows: fresh > HAD 40 °C > MVD 210 W > MVD 260 W > HAD 60 °C > HAD 80 °C > MVD 160 W > FD. The highest ergothioneine content after the aqueous extraction of fresh *P. ostreatus* mushrooms was found in the aqueous extract of mushrooms dried by hot-air at 40 °C; this drying condition best preserves the ergothioneine content compared to that of fresh mushrooms, presumably due to the low drying temperature, which is close to that of natural-air-drying [53], compared to the other hot-air-drying temperatures. For mushrooms dried by microwave-vacuum, a better preservation of ergothioneine content is observed for microwave powers at 210 W and 260 W at 20 kPa. This is presumably due to the rapid loss of moisture resulting from the microwave power-to-vacuum pressure ratio [17], which allowed for better structural quality, as seen in the SEM images, especially at the microwave power of 210 W at 20 kPa, showing no significant differences compared to the results for hot-air-drying at 40 °C in terms of ergothioneine retention. The retention of ergothioneine content in the extract of mushrooms dried by microwave-vacuum at 260 W/20 kPa is not significantly different from that at 210 W/20 kPa, despite the collapse of the structure seen in the SEM images. This was likely favored by the reduced boiling point due to the incorporation of a vacuum, allowing for the stability of ergothioneine, contrary to what occurred in the hot-air-drying of *Pleurotus* mushrooms at drying temperatures equal to or greater than 60 °C [54]. The influence of other factors present in the food matrix on the stability of ergothioneine should not be overlooked [55], as observed in the extract of mushrooms dried at 160 W and by freeze-drying. For the latter, the results for the ergothioneine content were similar to those obtained in previous studies for the extracts of *P. ostreatus* mushrooms dried by freeze-drying [6].

## 4. Conclusions

Drying is an effective method to preserve edible mushrooms, which are highly perishable due to their moisture content. Edible mushrooms are a recognized source of bioactive compounds, and drying treatments should aim to preserve them. In this study, it was observed that drying methods significantly influenced the physicochemical characteristics and extraction results of the matrix of the mushroom *P. ostreatus*. The physicochemical properties of dried *P. ostreatus* were evaluated through SEM images, rehydration rate, and color parameters. The SEM images showed that the effect of drying processes on the preservation of the structure and porosity depended on the drying treatment applied. Among all drying treatments, mushrooms dried by freeze-drying, hot-air-drying at 40 °C, and microwave-vacuum-drying at 210 W/20 kPa showed the best preservation. In terms of rehydration rate, considering all drying treatments, the predominant rehydration mechanism was fiber relaxation over pore diffusion, and the rehydration rate was consistent with the structural quality observed through SEM images. In terms of color parameters, a quality parameter for edible mushrooms is a higher L* value combined with a lower color difference (ΔE), and the mushrooms that achieved a better L* and ΔE ratio were those dried by freeze-drying, hot-air-drying at 40 °C, and microwave-vacuum-drying at 260 W/20 kPa. The influence of the drying treatments had a characteristic effect on the physicochemical characteristics of the dried mushrooms, also with an impact on the antioxidant properties, showing interesting trends. The evaluation of the antioxidant properties of extracts from fresh and dried *P. ostreatus* mushrooms included the assessment of phenolic compound content, antioxidant activity, and ergothioneine content, given the current interest in the antioxidant potential of edible mushrooms. Considering the physicochemical characteristics evaluated so far, the extracts from fresh mushrooms were the best in preserving the antioxidant properties, while those dried by hot-air-drying at 40 °C and microwave-vacuum-drying at 210 W/20 kPa and 260 W/20 kPa generally performed better. However, given the structural quality evaluated, microwave-vacuum-drying at 260 W/20 kPa would not be an ideal alternative. Thus, the drying treatments influenced the antioxidant properties evaluated by the extracts, making the optimization of extraction variables in fresh mushrooms a valuable reference for maximizing the content of bioactive compounds. Finally, among the drying treatments applied to the mushroom *P. ostreatus*, hot-air-drying at 40 °C, a traditional drying method, showed good physicochemical characteristics, preserving well the phenolic and ergothioneine contents, in addition to better maintaining their antioxidant activity, evaluated through different chemical reactions. Further research on edible mushrooms is necessary to understand the influence of drying processes on the matrix and the methods for extraction of bioactive compounds, especially regarding phenolic content and ergothioneine, which show interesting antioxidant activity.

## Figures and Tables

**Figure 1 antioxidants-13-01581-f001:**
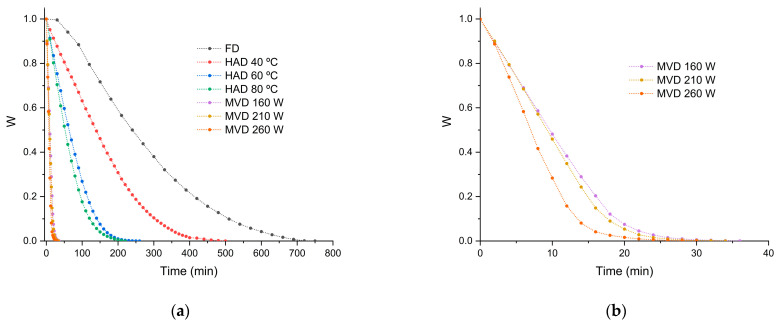
Moisture content curves of *P. ostreatus* mushroom dried by (**a**) different drying processes (FD, HAD, MVD) and in detail by (**b**) microwave-vacuum-drying at a vacuum pressure of 20 kPa for the three microwave powers.

**Figure 2 antioxidants-13-01581-f002:**
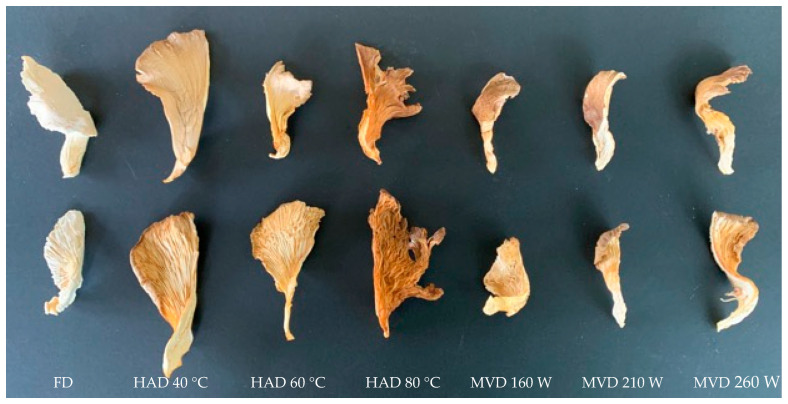
Macroscopic photos of the fungus *P. ostreatus*; from left to right, freeze-dried (FD); hot-air-dried (HAD) at 40, 60, and 80 °C; and microwave-vacuum-dried (MVD) at 160, 210, and 260 W at a vacuum pressure of 20 kPa.

**Figure 3 antioxidants-13-01581-f003:**
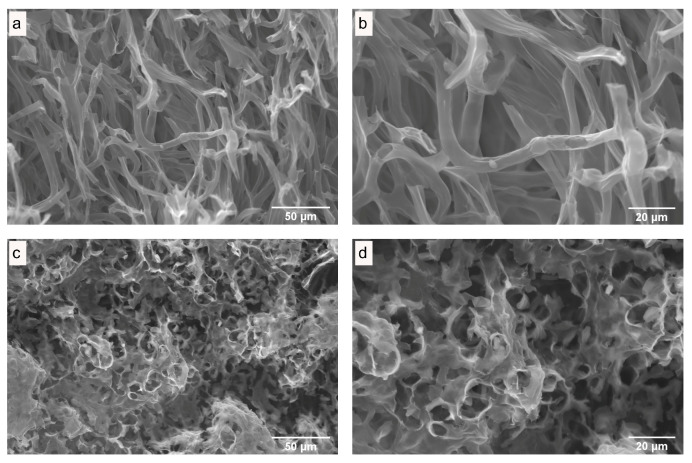
(**a**,**b**) Internal structure and (**c**,**d**) surface of freeze-dried *P. ostreatus*, as observed by SEM, at magnifications of (**a**,**b**) 1000X and (**c**,**d**) 2000X.

**Figure 4 antioxidants-13-01581-f004:**
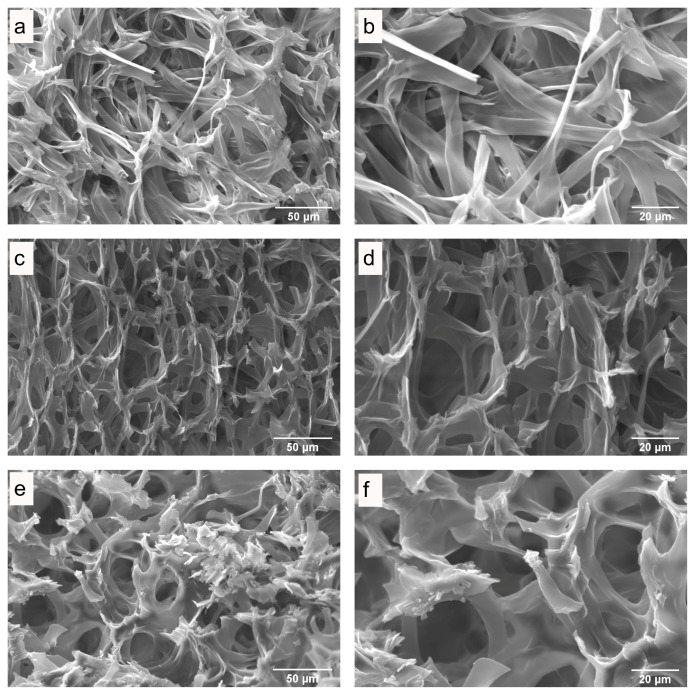
Internal structure of *P. ostreatus* dried by hot-air at temperatures of (**a**,**b**) 40 °C, (**c**,**d**) 60 °C, and (**e**,**f**) 80 °C, as obtained through SEM, at magnifications of (**a**,**c**,**e**) 1000X and (**b**,**d**,**f**) 2000X.

**Figure 5 antioxidants-13-01581-f005:**
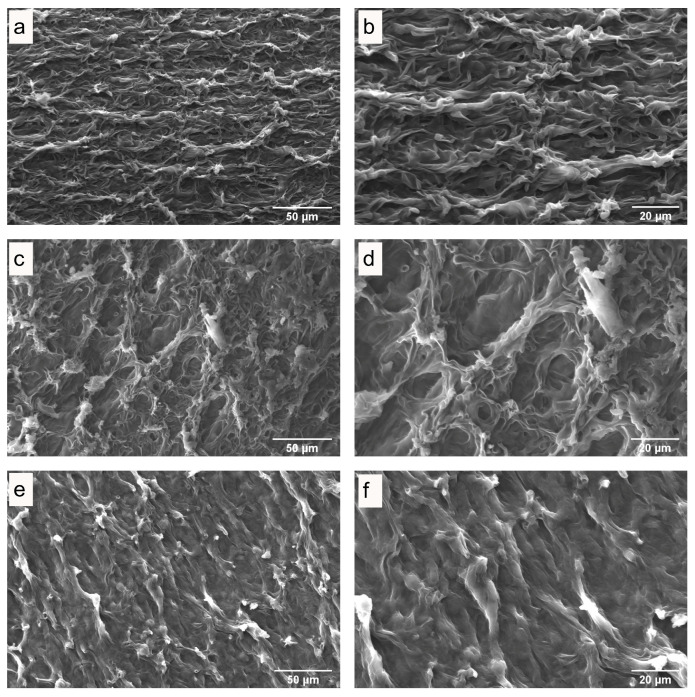
Surface of *P. ostreatus* dried by hot-air at temperatures of (**a**,**b**) 40 °C, (**c**,**d**) 60 °C, and (**e**,**f**) 80 °C, as obtained through SEM, at magnifications of (**a**,**c**,**e**) 1000X and (**b**,**d**,**f**) 2000X.

**Figure 6 antioxidants-13-01581-f006:**
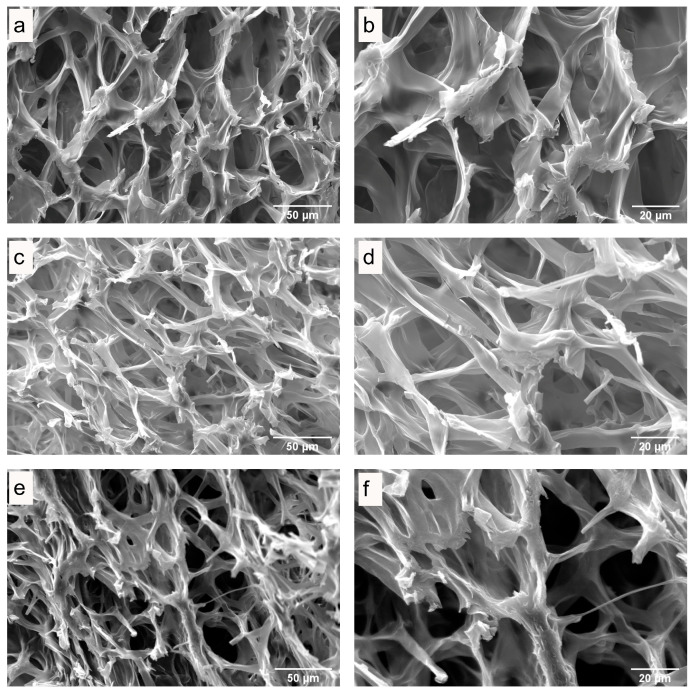
Internal structure of *P. ostreatus* dried by microwave-vacuum at a microwave power of (**a**,**b**) 160 W, (**c**,**d**) 210 W, and (**e**,**f**) 260 W at a vacuum pressure of 20 kPa, as observed through SEM, at magnifications of (**a**,**c**,**e**) 1000X and (**b**,**d**,**f**) 2000X.

**Figure 7 antioxidants-13-01581-f007:**
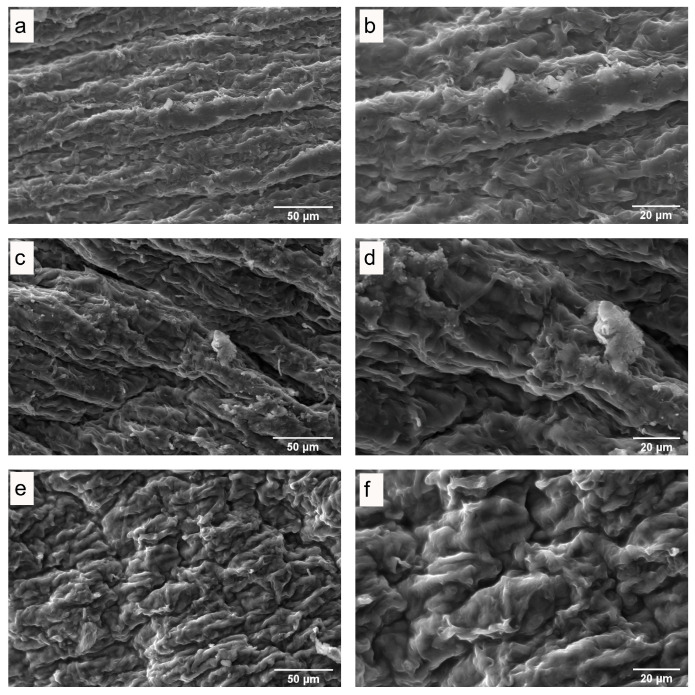
Surface of *P. ostreatus* dried by microwave-vacuum at a microwave power of (**a**,**b**) 160 W, (**c**,**d**) 210 W, and (**e**,**f**) 260 W at a vacuum pressure of 20 kPa, as observed through SEM, at magnifications of (**a**,**c**,**e**) 1000X and (**b**,**d**,**f**) 2000X.

**Figure 8 antioxidants-13-01581-f008:**
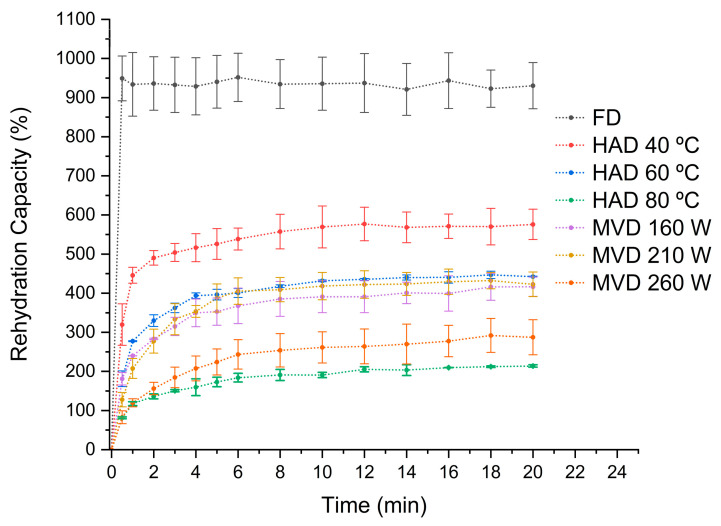
Rehydration rate curves of *P. ostreatus* mushrooms dried by freeze-drying (FD), hot-air-drying (HAD), and microwave-vacuum-drying (MVD) for different amounts of time.

**Figure 9 antioxidants-13-01581-f009:**
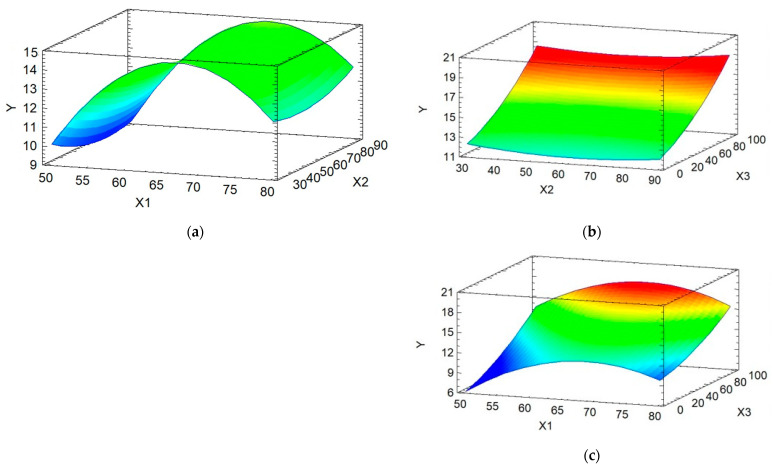
Response surface plots for total soluble phenolic content (Y) (mg GAE/g d.m.) of the factors and their interactions at (**a**) temperature (X_1_) (°C) and time (X_2_) (min); (**b**) time (X_2_) (min) and solvent (X_3_) (%); and (**c**) temperature (X_1_) (°C) and solvent (X_3_) (%).

**Figure 10 antioxidants-13-01581-f010:**
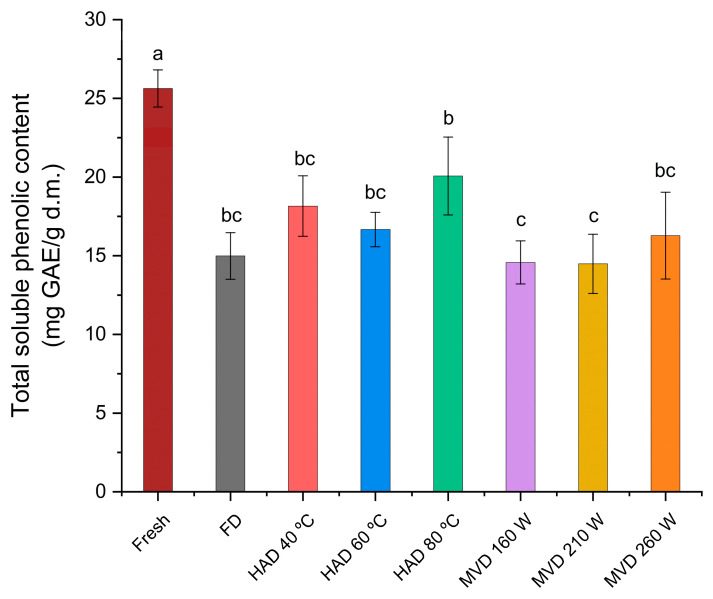
Total soluble phenolic content (mg GAE/g d.m.) of the aqueous extracts of fresh *P. ostreatus* mushroom and those prepared by freeze-drying (FD), hot-air-drying (HAD), and microwave-vacuum-drying (MVD). The results are expressed in dry mass (d.m.). Different letters indicate a significant difference (*p* ≤ 0.05), according to Tukey’s test.

**Figure 11 antioxidants-13-01581-f011:**
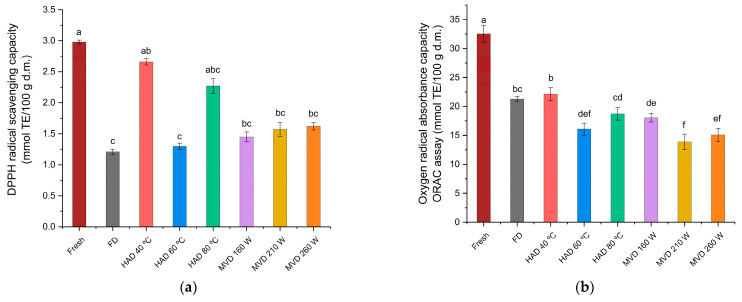
Antioxidant activity of aqueous extracts from fresh and dried *P. ostreatus* mushrooms evaluated through (**a**) DPPH radical capture capacity (mmol TE/100 g d.m.) and (**b**) oxygen radical absorption capacity (ORAC) (mmol TE/100 g d.m.) assays. The results are expressed in dry mass (d.m.). Different letters indicate a significant difference (*p* ≤ 0.05), according to Tukey’s test.

**Figure 12 antioxidants-13-01581-f012:**
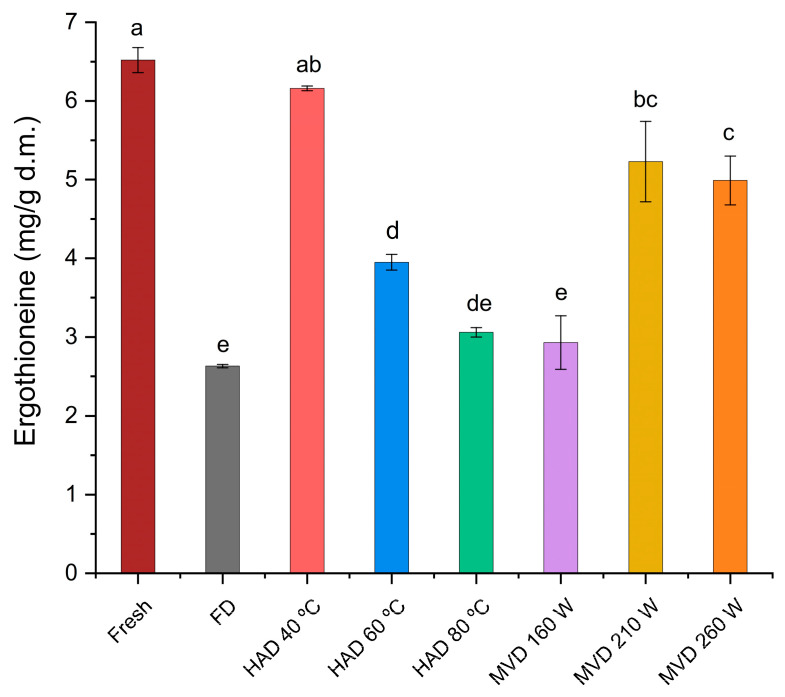
Ergothioneine content (mg/g d.m.) of the aqueous extracts of fresh *P. ostreatus* mushroom and prepared by freeze-drying (FD), hot-air-drying (HAD), and microwave-vacuum-drying (MVD). The results are expressed in dry mass (d.m.). Different letters indicate a significant difference (*p* ≤ 0.05), according to Tukey’s test.

**Table 1 antioxidants-13-01581-t001:** Fresh and dried *P. ostreatus* mushrooms and their color parameters.

Conservation Status	L*	a*	b*	ΔE
Fresh	51.23 ± 0.83 ^f^	−0.86 ± 0.04 ^a^	11.32 ± 0.33 ^b^	52.48 ± 0.89 ^e^
FD	46.88 ± 0.82 ^e^	−1.34 ± 0.15 ^a^	15.15 ± 0.32 ^c^	49.29 ± 0.68 ^d^
HAD 40 °C	23.74 ± 1.22 ^d^	2.45 ± 0.03 ^e^	16.76 ± 0.54 ^d^	29.16 ± 1.28 ^c^
HAD 60 °C	15.21 ± 0.49 ^b^	1.81 ± 0.11 ^cd^	11.02 ± 0.29 ^b^	18.87 ± 0.58 ^a^
HAD 80 °C	12.22 ± 0.95 ^a^	3.98 ± 0.09 ^f^	11.44 ± 0.39 ^b^	17.22 ± 0.92 ^a^
MVD 160 W	18.07 ± 0.30 ^c^	1.99 ± 0.06 ^d^	15.64 ± 0.05 ^cd^	23.98 ± 0.26 ^b^
MVD 210 W	15.76 ± 0.59 ^bc^	1.48 ± 0.37 ^bc^	11.13 ± 0.91 ^b^	19.36 ± 1.03 ^a^
MVD 260 W	21.61 ± 1.12 ^d^	1.22 ± 0.18 ^b^	8.97 ± 0.27 ^a^	23.43 ± 1.14 ^b^

Values are given as mean ± standard deviation of triplicate determinations. Different superscript letters indicate statistically significant differences between means (*p* < 0.05) for each parameter.

**Table 2 antioxidants-13-01581-t002:** Rehydration kinetic parameters.

Drying Processes	Peppas		
	k	n	R^2^
FD			
HAD 40 °C	0.7476	0.40	97.46
HAD 60 °C	0.6098	0.55	99.24
HAD 80 °C	0.5344	0.33	89.80
MVD 160 W	0.5920	0.40	99.97
MVD 210 W	0.4869	0.70	99.97
MVD 260 W	0.4348	0.46	94.65

**Table 3 antioxidants-13-01581-t003:** Total soluble phenolic content of fresh *P. ostreatus* mushroom for the different combinations of extraction conditions based on the three-level factorial experimental design in three blocks, with one central point per block, according to the response surface methodology (RSM) evaluation.

Run ^a^	Block	Temperature (X_1_) (°C)	Time (X_2_) (min)	Solvent (X_3_) (%)	Total Soluble Phenolic Content ^b^ (Y) (mg GAE/g d.m.)
1	1	65	60	0	11.80 ± 1.11
2	1	65	30	50	15.42 ± 1.68
3	1	50	90	50	8.87 ± 1.45
4	1	50	30	0	7.04 ± 0.43
5	1	80	90	0	10.20 ± 1.73
6	1	80	60	50	11.66 ± 2.33
7	1	80	30	100	14.89 ± 1.59
8	1	50	60	100	12.87 ± 1.59
9	1	65	90	100	19.03 ± 3.11
10	1	65	60	50	13.21 ± 1.21
11	2	65	90	50	14.04 ± 1.27
12	2	65	60	50	15.32 ± 2.87
13	2	80	60	0	10.10 ± 2.26
14	2	50	30	100	14.12 ± 0.35
15	2	50	60	50	9.26 ± 1.90
16	2	80	90	100	16.46 ± 1.59
17	2	50	90	0	6.71 ± 1.61
18	2	65	60	100	19.08 ± 1.68
19	2	65	30	0	11.68 ± 0.70
20	2	80	30	50	11.71 ± 1.73
21	3	65	90	0	11.46 ± 2.12
22	3	50	30	50	10.21 ± 1.63
23	3	65	60	50	12.37 ± 2.20
24	3	80	30	0	10.05 ± 1.44
25	3	80	60	100	15.72 ± 0.68
26	3	65	60	50	14.30 ± 1.60
27	3	65	30	100	19.51 ± 1.81
28	3	80	90	50	13.45 ± 1.35
29	3	50	90	100	14.42 ± 0.93
30	3	50	60	0	6.03 ± 1.17

^a^ Order of execution. ^b^ Values are given as mean ± standard deviation of triplicate determinations; total soluble phenolic content expressed as mg GAE/g d.m. d.m.—dry mass.

**Table 4 antioxidants-13-01581-t004:** One-way ANOVA of the independent variables (X_1_, X_2_, X_3_) for total soluble phenolic content.

Variables	*p*—Value
X_1_: Temperature (°C)	0.0000 *
X_2_: Time (min)	0.9976
X_3_: Solvent (%)	0.0000 *
X_1_^2^	0.0000 *
X_1_X_2_	0.0837
X_1_X_3_	0.0791
X_2_^2^	0.0691
X_2_X_3_	0.5060
X_3_^2^	0.0073 *

* Statistically significant differences (*p* < 0.05).

## Data Availability

All data and materials are available upon request from the corresponding author. The data are not publicly available due to ongoing research using part of the data.

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
