# Peer review of "Optimization of Phenolic Content Extraction and Effects of Drying Treatments on Physicochemical Characteristics and Antioxidant Properties of Edible Mushroom Pleurotus ostreatus (Jacq.) P. Kumm (Oyster Mushroom)"

_antioxidants, 2024, doi:10.3390/antiox13121581_

Round 1
Reviewer 1 Report
This study investigates the effects of different drying methods on mushrooms. While the results reveal some interesting trends, the novelty of this work needs to be better articulated. Additionally, the discussion section requires substantial improvement, particularly in explaining how and why various drying methods yielded different outcomes. This should include an analysis of correlations and the potential structure-function relationships underlying the observations.
Line 79: Examples of individual phenolics should be provided.
Lines 84-89: The manuscript title and the stated objectives are not well-aligned.
Section 2.1: Why were the samples not harvested directly from the field? Clarification is needed regarding the sourcing of materials.
Section 2.3: Which colorimeter was used?
Line 156: Which solvent?
Line 166: Do these temperatures maintain the thermosensitive compounds?
Section 2.8: Why were only soluble phenolics determined? How about insoluble-bound phenolics?
Line 494 and elsewhere: "TE" instead of "ET"
Section 3.8: Why does DPPH radical scavenging property show a very low level of activity?
See major comments
Author Response
Mr. Gowri
E-Mail: gowri.v@mdpi.com
Antioxidants
Manuscript: antioxidants-3346334
Dear Dr. Gowri
We are pleased to resubmit the revised version of our manuscript “Optimization of phenolic content extraction and effects of drying treatments on physicochemical characteristics and antioxidant properties of edible mushroom Pleurotus ostreatus (Jacq.) P. Kumm (Oyster mushroom).” for further consideration to publication in Antioxidants
In our response to reviewer’s comments, we described the changes introduced in the manuscript, where they were highlighted. Comments of reviewers were answered point by point, and we hope to have met the requirement of the journal.
Sincerely yours,
Dr. Ociel Muñoz-Fariña
Review 1
This study investigates the effects of different drying methods on mushrooms. While the results reveal some interesting trends, the novelty of this work needs to be better articulated. Additionally, the discussion section requires substantial improvement, particularly in explaining how and why various drying methods yielded different outcomes. This should include an analysis of correlations and the potential structure-function relationships underlying the observations.
Line 79: Examples of individual phenolics should be provided.
- Thanks for the reviewer's suggestions: Line 77-80 the sentence. “In edible mushrooms, considerable efforts have been made to find effective extraction methods that have better efficiency and yield extracts endowed with high bioactivity, particularly concerning phenolic compounds due to their close relationship with antioxidant activity [20],” has been replaced by “In edible mushrooms, considerable efforts have been made to find effective extraction methods with higher efficiency and yielding extracts with high bioactivity, especially with respect to phenolic compounds due to their close relationship with antioxidant activity [22], especially in phenolic acids, the main phenolic compounds in mushrooms [5][23], and cinnamic acid, p-coumaric acid, p-hydroxybenzoic acid, and chlorogenic acid stand out in P. ostreatus [24]” (see line 85 -90)
Lines 84-89: The manuscript title and the stated objectives are not well-aligned.
- Thanks for the reviewer's suggestions. The title “Effects of drying treatments on the physicochemical character-istics of the edible mushroom Pleurotus ostreatus (Jacq.) P. Kumm (Oyster mushroom)” has been replaced by “Optimization of phenolic content extraction and effects of drying treatments on physicochemical characteristics and anti-oxidant properties of edible mushroom Pleurotus ostreatus (Jacq.) P. Kumm (Oyster mushroom).”
Section 2.1: Why were the samples not harvested directly from the field? Clarification is needed regarding the sourcing of materials.
- We thank the reviewers for this question. The research was based on the cultivable and commercial potential of Pleurotus ostreatus (oyster mushroom). It is a mushroom that is available all year round in the local trade, in a context where it is difficult to harvest due to its limited presence in the forests of southern Chile. With our research, we want to help lay the foundations for applied research on this food, with an interest in biomedical research, using extracts rich in phenolic content and ergothioneine.
Section 2.3: Which colorimeter was used?
- We thank the reviewers for the question. lines 127-129 The following paragraph was modified “The color measurement of fresh and dried P. ostreatus samples was determined as described by Hou et al. [30]. The results were expressed as CIE Lab values. The assay was performed in triplicate” has been replaced by “The color measurement of fresh and dried P. ostreatus samples was determined as described by Hou et al. [30]. Color parameters of the samples were evaluated by reflectance with a colorimeter (CR-400, Minolta Chroma Meter, Tokyo, Japan) that was previously calibrated with a blank. The results were expressed as CIE Lab values. The assay was performed in triplicate”. (see lines 138-142)
Line 156: Which solvent?
- Thanks for the reviewer's suggestions. Line 156-157 the sentence “Subsequently, 15 mL of solvent was added and manually homogenized and extracted in an ultrasonic water bath” has been replaced by “Subsequently, 15 mL of an aqueous solvent (in different ratios v/v ethanol/distilled water of 100/0, 50/50 and 0/100 in each assay) was added and manually homogenized and extracted in an ultrasonic water bath at 67 ± 2 ºC”.(see line 162-171)
Line 166: Do these temperatures maintain the thermosensitive compounds?
- Thank you for the question from the reviewer. Through the response surface model and considering the factors of each variable, for the extraction of the compounds of interest, it is possible to observe that at temperatures above 70 ºC of extraction, the retention of phenolic content in the extracts decreases, presumably due to thermosensitivity, in addition to other factors such as the interaction of solvent, temperature and fungal matrix.
Section 2.8: Why were only soluble phenolics determined? How about insoluble-bound phenolics?
- The aim of this study was to optimize the extraction and quantification of phenolic compounds, specifically focusing on soluble phenolic compounds, from oyster mushrooms. However, as noted by the reviewer, phenolic compounds bound to other structures within the mushroom, which are not extractable, may also be present—similar to those found in berries. These bound compounds, however, were not the focus of this study. Therefore, we emphasize that our analysis is centered on soluble phenolic compounds.
Line 494 and elsewhere: "TE" instead of "ET"
- Thank you for this comment from the reviewers. We have thoroughly reviewed the manuscript, and the errors have been corrected.
Section 3.8: Why does DPPH radical scavenging property show a very low level of activity?
- Thank you for the reviewer's comment. We have checked and had a typo in the unit of measurement. We have edited. The unit of measure “moles ET/100 d.m.” was replaced by “mmoles TE/100 d.m.”.
Reviewer 2 Report
This manuscript, Effects of drying treatments on the physicochemical characteristics of the edible mushroom Pleurotus ostreatus (Jacq.) P. Kumm (Oyster mushroom), investigated the effects of different drying methods on the physicochemical properties of P. ostreatus. The mushrooms were subjected to freeze-drying, hot-air drying and microwave vacuum drying, that were then assessed through rehydration capacity, color, microstructural properties by scanning electron microscopy, as well as total phenolic content, antioxidant activity determined by DPPH and ORAC assays, and ergothioneine levels of the fresh and dried P. ostreatus extracts. The experimental design method is reasonable, results and data interpretation could well support this study. I believe that the manuscript contains interesting studies for readers. My suggestion is that they need to add major revision as commented below.
1. The abstract would benefit from the results and a concluding remark. Also, the focus of the first section should be on the importance and novelty of this manuscript than the general information about edible mushrooms.
2. The study objective and the scientific problem need to be further condensed in the Introduction section.
1. Lines 101-102: constant weight? What was the standard?
2. Lines 500-502: these explanations and discussions are highly ambiguous. Please analyze the reason deeply combined with the results of total soluble phenolic content.
3. Lines 545-568: the conclusion part needs to reorganize the language to enrich and sublimate the research results.
Author Response
Mr. Gowri
E-Mail: gowri.v@mdpi.com
Antioxidants
Manuscript: antioxidants-3346334
Dear Dr. Gowri
We are pleased to resubmit the revised version of our manuscript “Optimization of phenolic content extraction and effects of drying treatments on physicochemical characteristics and antioxidant properties of edible mushroom Pleurotus ostreatus (Jacq.) P. Kumm (Oyster mushroom).” for further consideration to publication in Antioxidants
In our response to reviewer’s comments, we described the changes introduced in the manuscript, where they were highlighted. Comments of reviewers were answered point by point, and we hope to have met the requirement of the journal.
Sincerely yours,
Dr. Ociel Muñoz-Fariña
Review 2
This manuscript, Effects of drying treatments on the physicochemical characteristics of the edible mushroom Pleurotus ostreatus (Jacq.) P. Kumm (Oyster mushroom), investigated the effects of different drying methods on the physicochemical properties of P. ostreatus. The mushrooms were subjected to freeze-drying, hot-air drying and microwave vacuum drying, that were then assessed through rehydration capacity, color, microstructural properties by scanning electron microscopy, as well as total phenolic content, antioxidant activity determined by DPPH and ORAC assays, and ergothioneine levels of the fresh and dried P. ostreatus extracts. The experimental design method is reasonable, results and data interpretation could well support this study. I believe that the manuscript contains interesting studies for readers. My suggestion is that they need to add major revision as commented below.
- The abstract would benefit from the results and a concluding remark. Also, the focus of the first section should be on the importance and novelty of this manuscript than the general information about edible mushrooms.
- The study objective and the scientific problem need to be further condensed in the Introduction section.
- Thank for the suggestion to the reviewer. We have included more elements in the development of the overall research.
- Lines 101-102: constant weight? What was the standard?
- We do not fully understand the nature of your inquiry; however, we can clarify that gravimetric analyses for determining moisture content are conducted until weight loss ceases under the applied thermal treatment. That is, until a constant weight is achieved and indicates that the drying treatment has been completed.".
- Lines 500-502: these explanations and discussions are highly ambiguous. Please analyze the reason deeply combined with the results of total soluble phenolic content.
- Thank you for the reviewer's comment. Line 500-502 the sentence “The antioxidant activity of the aqueous extracts from fresh and dried P. ostreatus may be related to phenolic composition, as they are the main carriers of antioxidants [6][23]” has been replaced by “The phenolic content of edible mushrooms serves as one of the most effective antioxidants [50] and are the main carriers of antioxidants; they can donate electrons and hydrogen atoms, it has been observed that aqueous extracts of P. ostreatus could scavenge DPPH radicals and absorb oxygen radicals (ORAC) [26]. The trend observed between total phenolic content and antioxidant activity through DPPH and ORAC assays of aqueous extracts of P. ostreatus is consistent with previous research on extracts of edible mushrooms, where a higher amount of phenolic content has been associated with increased antioxidant activity [6][51][52]. In particular, the results obtained with the aqueous extracts of P. os-treatus dried by hot-air at 80 ºC in the DPPH assay (Figure 10a) can be attributed to the action of other compounds in addition to the phenolic content [52]. In the case of the ORAC assay in general, the trend observed between the total phenolic content (Figure 10) and the assay results (Figure 11b) is in line with research on edible mushrooms, as a good correlation has generally been found [50]. It is important to mention that the antioxidant activity of P. ostreatus extracts can be attributed to the presence of many bioactive compounds, including flavonoids, phenols and peptides [47], given the complexity of the chemical composition of P. ostreatus mushrooms [26]. It has even been observed that each compound within the phenolic content may have different antioxidant activity [53]. Nevertheless, a significant contribution of the phenolic content to the antioxidant properties has been observed in edible mushrooms, including P. ostreatus, which could also explain the relationship between phenolic content and the antioxidant activity evaluated [51]”.(see Line 532-552)
- Lines 545-568: the conclusion part needs to reorganize the language to enrich and sublimate the research results.
- We are grateful for the reviewer's guidance. The conclusion has been redrafted. (see lines 598 – 633)
Round 2
Reviewer 1 Report
The authors have now revised the manuscript accordingly, and it is ready to move to the next steps. The only comment is I could not find these lines “In edible mushrooms, considerable efforts have been made to find effective extraction methods with higher efficiency and yielding extracts with high bioactivity, especially with respect to phenolic compounds due to their close relationship with antioxidant activity [22], especially in phenolic acids, the main phenolic compounds in mushrooms [5][23], and cinnamic acid, p-coumaric acid, p-hydroxybenzoic acid, and chlorogenic acid stand out in P. ostreatus [24]” (see line 85 -90)" on the manuscript.
N/A
Reviewer 2 Report
ok
ok